# Lessons Learned and Future Perspectives for Rotavirus Vaccines Switch in the World Health Organization, Regional Office for Africa

**DOI:** 10.3390/vaccines11040788

**Published:** 2023-04-03

**Authors:** Inacio Mandomando, Joseph Nsiari-Muzeyi Biey, Gilson Paluku, Mutale Mumba, Jason M. Mwenda

**Affiliations:** 1Centro de Investigação em Saúde de Manhiça (CISM), Maputo P.O. Box 1929, Mozambique; 2Instituto Nacional de Saúde (INS), Maputo P.O. Box 3943, Mozambique; 3ISGlobal, Hospital Clínic, Universitat de Barcelona, 08036 Barcelona, Spain; 4Inter Country Support Team (IST) for West Africa, Regional Office for Africa, World Health Organization (WHO), Ouagadougou 03 BP 7019, Burkina Faso; 5Inter Country Support Team (IST) for Central Africa, World Health Organization, Libreville P.O. Box 820, Gabon; 6Inter Country Support Team (IST) for East and Southern Africa, Regional Office for Africa, World Health Organization, Harare P.O. Box 5160, Zimbabwe; 7Regional Office for Africa, World Health Organization (WHO), Brazzaville P.O. Box 06, Congo

**Keywords:** rotavirus, vaccine, switch, WHO AFR

## Abstract

Background: Following the World Health Organization (WHO) recommendation, 38/47 countries have introduced rotavirus vaccines into the program of immunization in the WHO Regional Office for Africa (WHO/AFRO). Initially, two vaccines (Rotarix and Rotateq) were recommended and recently two additional vaccines (Rotavac and Rotasiil) have become available. However, the global supply challenges have increasingly forced some countries in Africa to switch vaccine products. Therefore, the recent WHO pre-qualified vaccines (Rotavac, Rotasiil) manufactured in India, offer alternatives and reduce global supply challenges related to rotavirus vaccines; Methods: Using a questionnaire, we administered to the Program Managers, Expanded Program for Immunization, we collected data on vaccine introduction and vaccine switch and the key drivers of the decisions for switching vaccines products, in the WHO/AFRO. Data was also collected fromliterature review and the global new vaccine introduction status data base maintained by WHO and other agencies. Results: Of the 38 countries that introduced the vaccine, 35 (92%) initially adopted Rotateq or Rotarix; and 23% (8/35) switched between products after rotavirus vaccine introduction to either Rotavac (n = 3), Rotasiil (n = 2) or Rotarix (n = 3). Three countries (Benin, Democratic Republic of Congo and Nigeria) introduced the rotavirus vaccines manufactured in India. The decision to either introduce or switch to the Indian vaccines was predominately driven by global supply challenges or supply shortage. The withdrawal of Rotateq from the African market, or cost-saving for countries that graduated or in transition from Gavi support was another reason to switch the vaccine; Conclusions: The recently WHO pre-qualified vaccines have offered the countries, opportunities to adopt these cost-effective products, particularly for countries that have graduated or transitioning from full Gavi support, to sustain the demand of vaccines products.

## 1. Introduction

Since the World Health Organization (WHO) recommendation for the global introduction of rotavirus vaccine in high disease burden countries [1], Gavi, the Vaccine Alliance, has provided funding for the introduction of rotavirus vaccine in 53 Gavi-eligible countries globally [2]. The WHO has been providing technical guidance and support to the African countries to enable the countries make evidence based decision on the introduction of rotavirus vaccines [3]. In sub-Saharan Africa, 38/47 (80%) of the countries introduced the rotavirus vaccines in their National Immunization Programs [2], with Rotarix being the most adopted vaccine [4]. However, more recently, additional WHO pre-qualified rotavirus vaccines manufactured in India: Rotavac (Human neonatal strain 116E G9P [11]) and RotasIIL (Pentavalent human–bovine reassortant serotypes G (1–4) and G9) are increasingly being deployed in sub-Saharan Africa [5].

The early adopted rotavirus vaccines have shown beneficial impact on reducing childhood morbidity and -mortality due to rotavirus and cost-effective in low- and middle income countries [6,7,8,9,10]. However, vaccine cost, demand, manufacturer production capacity and programmatic issues are, among others, the major barriers to the long-term sustainability of rotavirus vaccination, particularly for countries scheduled to graduate from Gavi financial support. Also, local Government co-financing is required to meet operation costs at the country level. On the other hand, cost-effectiveness analysis comparing Rotarix, Rotateq and Rotavac showed lower price per dose for Rotavac vaccine compared to the competitor products [11], and this led countries like Palestine to switch from Rotarix to Rotavac vaccine after assuming the financial responsibility for rotavirus vaccine procurement [12].

The decisions influencing to switching from one vaccine product to another should systematically assess multiple criteria beyond vaccine price include the health, programmatic and economic aspects. Countries should also fully evaluate all product characteristics including product presentation, number of doses per course, cold chain volume, cost of delivery, and wastage [12]. WHO recommends that the choice of rotavirus vaccine product to be used in a country, either at first introduction or at switch, should be based on disease burden, programmatic characteristics, vaccine supply, and vaccine price including cost effectiveness analysis. Current evidence indicates that local data on circulating rotavirus strains should not drive product choice, as the WHO prequalified rotavirus vaccines have been shown to provide cross protection against heterologous strains [1].

A number of countries switched from Rotateq or Rotarix to Rotavac, Rotasiil but the reason for the decisions to switch vaccines are not yet fully documented. In this evaluation, we investigated the rotavirus vaccine being used in WHO/AFRO countries and assessed the reasons for switching vaccine products at a country level.

## 2. Materials and Methods

To assess rotavirus vaccines that are in use in WHO/AFRO countries that participate in the African Rotavirus Surveillance Network introduced in their immunization programs and whether these vaccines had changed since the introduction, we collected information by administering a questionnaire to the Program Managers, Expanded Program for Immunization of the countries that had switched or planning to switch the vaccine product. Data was also collected from literature review by searching the terms or combination of the terms “rotavirus vaccine switch in WHO AFRO” in the PubMed (https://pubmed.ncbi.nlm.nih.gov/, accessed on 15 February 2023), WHO (https://www.who.int/, accessed on 15 February 2023), International Vaccine Access Center [IVAC] (https://www.jhsph.edu/ivac/, accessed on 15 February 2023) websites. The data collected included the type of vaccine and if the country had changed or planned to change products since introduction. In addition, the data of current status of global rotavirus introductions was accessed [2]. Data that was collected was entered in the excel sheet, checked for consistency and analysis performed using Stata version 14 (StataCorp LP, College Station, TX, USA).

## 3. Results

As of 30 October 2022, 38 out of 47 (80%) of the countries in WHO African region had introduced rotavirus vaccines in their national immunization programs, with Nigeria being the most recent country that started phased vaccine introduction from August 2022. Of those, six (Eswatini, Botswana, Mauritius, Namibia, Seychelles and South Africa) are not-eligible for Gavi funding for vaccine introduction and three countries (Angola, Republic of Congo and Ghana) have graduated or transitioned, and in this case, national Governments must budget and allocate funds for the procurement of vaccines and operational costs for vaccine implementation. The remaining are Gavi Phase III eligible (Table 1).

Of the 38 countries that introduced the rotavirus vaccine in the WHO/AFRO, three (Benin, Democratic Republic of Congo and Nigeria) were first time introductions of the recently pre-qualified Indian manufactured vaccines, while the remaining 35 introduced Rotateq or Rotarix. In 2009, when rotavirus vaccination was being rolled out, only two vaccine products were available (Rotateq or Rotarix). However, from the first 35 group of countries to introduce, 23% (n = 8) of the countries have switched the initially adopted rotavirus vaccines to an alternate, recently prequalified Indian vaccines. Among these eight countries that switched, five (63%) adopted Indian manufactured vaccines, with Rotavac being the most frequently adopted (n = 3), while only two have adopted RotaSiiL. The rotavirus vaccines types used during initial introduction in the WHO/AFRO is shown in Table 2, including the list of countries that switched the initially adopted rotavirus vaccines and the factors driving the switch.

Six countries (Rwanda, The Gambia, Burkina Faso, Cote d’Ivore, Mali, Sao Tome and Principle) switched from Rotateq to other vaccines because the manufacturer withdrew this vaccine from African market due to undisclosed reasons. Tanzania was recently forced to switch due to the supply shortage. While Ghana’s decision to switch vaccine products was due to cost as the country graduated from GAVI support and had to self-fund the vaccine procurement (Table 2).

## 4. Discussion

In the last decade, we have witnessed the widespread adoption of the two second generation of WHO pre-qualified rotavirus vaccines, internationally licensed for preventing severe illness due to rotavirus diarrhea in high mortality burden countries, particularly in sub-Saharan Africa. The rate of rotavirus vaccine introduction was driven by financial support from Gavi as almost 84% (32/38) of the countries that introduced rotavirus vaccine in their EPI in the WHO/AFRO, did so with Gavi financial support. However, the WHO prequalification of additional two rotavirus vaccines: ROTAVAC™ (Bharat Biotech, Hyderabad, India) and RotaSIIL (Serum Institute of India, Pune, India) in 2018 [1], offers countries that have not yet introduced the vaccines wider vaccine product choices and pricing [13]. In addition, the competitive market safeguards reasonable pricing and aids countries in ensuring long-term sustainability of their vaccine programs as countries graduate from Gavi support.

These newer vaccines also provided additional options for countries that previously introduced the vaccines to consider more cost-effective vaccines [14]. In the WHO/AFRO, so far eight countries including one that have graduated from Gavi eligibility (Ghana) have switched the vaccines. The major factors that complicated this switch included the programmatic factors including (e.g., insufficient cold-chain capacity); the lack of a country-specific cost effectiveness data, anticipated Gavi transition and global supply shortages (personal communication EPI Managers). Interestingly, the initial switch was driven either by manufacturer removal of Rotateq vaccine or the cost-saving on operation costs/co-financing for countries that introduced this vaccine [15]. For example, the switch in Ghana was necessitated by anticipated Gavi transition and the recommendation was supported and endorsed by National Immunization Technical Advisory Group (NITAG) [16]; while Rwanda, The Gambia, Burkina Faso, Cote d’Ivore, Mali, Sao Tome and Principle were forced to switch because of the removal of Rotateq from the African market. The sustainability of rotavirus vaccines production is a matter of concern. Currently, some countries are facing supply shortages of their chosen product, Tanzania was the most recent country to switch to the Indian-manufactured vaccines, Rotavac

As many countries will transition from Gavi-eligibility in future, more countries will have to find additional resources to procure vaccines and fund their vaccine programs potentially necessitating a switch to a lower cost product. Notably, previous rotavirus vaccine switch that necessitated, the countries were informed about the decision to switch vaccine, meant country had adequate time to plan and manage the switch. More recently, five countries including Kenya, Senegal, Tanzania, Zambia and Zimbabwe have been informed by Gavi to switch from Rotarix to Indian manufactured vaccines (RotasIIL or Rotavac) abruptly due to unforeseen supply constraints. Currently, there is uncertainty in sustainable supply of early adopted rotavirus vaccines for African countries. To mitigate this supply constraints for Rotarix, Uganda notified Gavi of the intention to switch to Indian-manufactured vaccines due to unforeseen supply constraints with their current vaccine product. It is anticipated this unpredictable vaccine supply may erode the gains and documented benefits of rotavirus vaccination in Africa. African governments should support the recent initiatives of vaccine manufacturing as agreed during the African ministerial conference on immunization and Addis declaration [17] to cushion against this unpredictable vaccine supply for African countries.

Rotateq vaccine was completely withdrawn and no longer in use in the African region and there is eminent global supply constraint for Rotarix vaccine with five African countries currently affected. Although, the recently WHO pre-qualified vaccines offer opportunities for the countries to adopt these cost-effective products with broader availability, particularly for countries in the trajectory and transitioning from full Gavi support, these challenges emphasizes the need for future vaccines locally manufactured to sustain the local demand while the countries with high birth cohort are rolling out the vaccines (e.g., Nigeria and Democratic Republic of Congo) will sustainable supply.

## 5. Conclusions

The recently WHO pre-qualified vaccines have offered opportunities for more options for the countries to adopt these cost-effective products, particularly for countries that have graduated or transitioning from full Gavi support. Sustained supply of vaccines should be maintained to ensure availability for all countries including those with high birth cohort to reap the full benefits of these vaccination.

## Figures and Tables

**Table 1 vaccines-11-00788-t001:** Gavi eligibility and World Bank Income grouping of the countries that introduced rotavirus vaccines (initial introduction) in the WHO/AFRO.

Country	RV Introduction Date	Vaccine Type	Current GAVI Eligibility	World Bank Income Group
Angola	2014-04-28	ROTARIX (RV1)	Graduated	Lower middle income
Benin	2019-12-01	ROTAVAC (RV1)	Gavi Phase III Eligible	Lower middle income
Botswana	2012-07-03	ROTARIX (RV1)	Not Eligible	Upper middle income
Burkina Faso	2013-10-31	ROTASIIL (RV5)	Gavi Phase III Eligible	Low income
Burundi	2013-12-16	ROTARIX (RV1)	Gavi Phase III Eligible	Low income
Cameroon	2014-03-28	ROTARIX (RV1)	Gavi Phase III Eligible	Lower middle income
Congo	2014-04-24	ROTARIX (RV1)	Graduated	Lower middle income
Congo, Democratic Republic of the	2019-10-01	ROTASIIL (RV5)	Gavi Phase III Eligible	Low income
Côte d’Ivoire	2017-03-01	RotaTeq (RV5)	Gavi Phase III Eligible	Lower middle income
Eritrea	2014-08-14	ROTARIX (RV1)	Gavi Phase III Eligible	Low income
Eswatini	2015-05-12	ROTARIX (RV1)	Not Eligible	Lower middle income
Ethiopia	2013-11-07	ROTARIX (RV1)	Gavi Phase III Eligible	Low income
Gambia	2013-08-14	ROTARIX (RV1)	Gavi Phase III Eligible	Low income
Ghana	2012-04-26	ROTAVAC (RV1)	Graduated	Lower middle income
Guinea-Bissau	2015-11-24	ROTARIX (RV1)	Gavi Phase III Eligible	Low income
Kenya	2014-07-01	ROTARIX (RV1)	Gavi Phase III Eligible	Lower middle income
Lesotho	2017-12-18	ROTARIX (RV1)	Gavi Phase III Eligible	Lower middle income
Liberia	2016-04-25	ROTARIX (RV1)	Gavi Phase III Eligible	Low income
Madagascar	2014-05-05	ROTARIX (RV1)	Gavi Phase III Eligible	Low income
Malawi	2012-10-29	ROTARIX (RV1)	Gavi Phase III Eligible	Low income
Mali	2014-01-14	RotaTeq (RV5)	Gavi Phase III Eligible	Low income
Mauritania	2014-12-06	ROTARIX (RV1)	Gavi Phase III Eligible	Lower middle income
Mauritius	2015-05-01	ROTARIX (RV1)	Not Eligible	Upper middle income
Mozambique	2015-09-04	ROTARIX (RV1)	Gavi Phase III Eligible	Low income
Namibia	2014-11-11	ROTARIX (RV1)	Not Eligible	Upper middle income
Niger	2014-08-05	ROTARIX (RV1)	Gavi Phase III Eligible	Low income
Nigeria	2022/08/22	ROTAVAC (RV1)	Graduated	Lower middle income
Rwanda	2012-05-25	ROTARIX (RV1)	Gavi Phase III Eligible	Low income
Sao Tome and Principe	2016-09-22	RotaTeq (RV5)	Gavi Phase III Eligible	Lower middle income
Senegal	2014-11-28	ROTARIX (RV1)	Gavi Phase III Eligible	Lower middle income
Seychelles	2017-09-01	ROTARIX (RV1)	Not Eligible	High income
Sierra Leone	2014-03-28	ROTARIX (RV1)	Gavi Phase III Eligible	Low income
South Africa	2009-08-01	ROTARIX (RV1)	Not Eligible	Upper middle income
United Republic of Tanzania	2012-12-06	ROTARIX (RV1)	Gavi Phase III Eligible	Lower middle income
Togo	2014-06-19	ROTARIX (RV1)	Gavi Phase III Eligible	Low income
Uganda	2018-06-26	ROTARIX (RV1)	Gavi Phase III Eligible	Low income
Zambia	2013-11-26	ROTARIX (RV1)	Gavi Phase III Eligible	Lower middle income
Zimbabwe	2014-05-01	ROTARIX (RV1)	Gavi Phase III Eligible	Lower middle income

**Table 2 vaccines-11-00788-t002:** Countries that have switched rotavirus vaccines in the WHO African region and the factors driven the decision.

Country	Vaccine	Date of Introduction	Date of Switch	New Vaccine	Reason for Switch
Burkina Faso	Rotateq	31 October 2013	January, 2019	RotaSIIL	1, 3
Cote d’Ivore	Rotateq	1 March 2017	May, 2019	Rotarix	1, 3
The Gambia	Rotateq	14 October 2013	February, 2017	Rotarix	1, 3
Ghana	Rotarix	26 April 2012	January, 2020	Rotavac	2
Mali	Rotateq	14 January 2014	January, 2020	RotaSIIL	1, 3
Rwanda	Rotateq	25 May 2012	April, 2017	Rotarix	1, 3
São Tomé e Principe	Rotateq	22 September 2016	2018/2019	Rotavac	1, 3
The Gambia, United Republic of Tanzania	Rotarix	6 December 2012	October, 2022	Rotavac	4

Reason for switch code: 1 Challenges with supply after Rotateq removed from GAVI market; 2 Cost factors associated with graduation from GAVI support; 3 Challenges in cold chain; 4 Supply shortages.

## Data Availability

Not applicable.

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
