# Peer review of "Lessons Learned and Future Perspectives for Rotavirus Vaccines Switch in the World Health Organization, Regional Office for Africa"

_vaccines, 2023, doi:10.3390/vaccines11040788_

Round 1

Reviewer 1 Report

This important brief report highlights the threat of inadequate vaccine supply in national rotavirus vaccination programmes and details the benefit of having additional WHO pre-qualified oral rotavirus vaccines available for use in the AFRO region. I have a few comments/clarifications as follows:

1.       General:

a.       Language – there are several instances where the grammar and sentence structure are not correct. The article would benefit from attention to this as some sentences/points are difficult to understand. Also be consistent with use of terms e.g. Gavi vs. GAVI. Examples - Sometimes a “the” or “to” is missing e.g. line 46: “enable countries make evidence based decision”; line 93: “with Nigeria being more recent country”; line 80 “WHO-AFR”; line 88 “Data was”. Line 135: “sustainability as the countries graduate to the GAVI eligibility.”

b.       References – review and update some of the references where the web page is not provided e.g. Refs 2 and 14.

2.       Abstract: conclusion needs to be rephrased. States “… opportunities for three options:…” but three options are not given? Meaning is unclear.

3.       Introduction: Line 44/45 – suggest giving the number of GAVI-eligible countries in AFRO upfront as the focus of the article is on this region. Include details of the different Indian vaccine formulations. 

4.       Methods: The study is only descriptive. I did not see any frequencies in the tables. The methods should state this rather than stating that “analysis performed using stata version 14 (StataCorp LP, College Station, Texas, USA), and summarized in tables of frequencies.” (line 89,90).

5.       Results:

a.       Table 1: Text states that Nigeria adopted the recently pre-qualified Indian manufactured vaccines (line 103,104) but Table 1 shows that Nigeria introduced Rotarix (RV1). Please clarify.

b.       Table 2: Line 106 states eight countries have switched to an alternate vaccine yet Table 2 shows 9 countries. Text states that Benin adopted the recently pre-qualified Indian manufactured vaccines (line 103,104) but Table 2 shows Benin with date of introduction and date of switch as Dec 2019 with switch from Rotateq to Rotavac. Please clarify.

c.       Table 2: Column 3 should be “Date of introduction” rather than “Year of introduction”. Column 4 (Has the country switched vaccine) can be removed as the table only shows countries that have switched.

d.       Table 2 is repeated – remove duplicate table.

e.       Data are not complete for Table 2 - Not all countries have “Reason for switch” and “Date of switch” in the table.  

f.        Table 3: Are there price differences for the different Rotavac and RotaSiil formulations? Which formulations have been used in the countries adopting the Indian vaccines?

g.       It would also be good to provide some additional details on the different Indian vaccine formulations and benefits of using one over another as well as availability (either in the introduction section or when describing Table 3 in the results).

6.       Discussion:

a.       Line 139-141 discusses factors complicating the switch in vaccines. Can this be shown in the results when presenting reasons for switching. Assuming these data were collected?

b.       Line 149-150: Is it possible to give an indication of supply available for the different vaccines in the results section e.g. in Table 3? From the discussion, the issue seems to be with supplies of Rotarix, but are there adequate supplies of the two Indian vaccines? Or would this require additional manufacturers?

7.       Conclusion – suggest revising this sentence as is very long and it’s difficult to understand the overall meaning.

Author Response

Reviewer 1

This important brief report highlights the threat of inadequate vaccine supply in national rotavirus vaccination programs and details the benefit of having additional WHO pre-qualified oral rotavirus vaccines available for use in the AFRO region. I have a few comments/clarifications as follows:

  1. General:
  2. Language – there are several instances where the grammar and sentence structure are not correct. The article would benefit from attention to this as some sentences/points are difficult to understand. Also be consistent with use of terms e.g. Gavi vs. GAVI. Examples - Sometimes a “the” or “to” is missing e.g. line 46: “enable countries make evidence based decision”; line 93: “with Nigeria being more recent country”; line 80 “WHO-AFR”; line 88 “Data was”. Line 135: “sustainability as the countries graduate to the GAVI eligibility.”

Response: We thank the reviewer for these observations. The sentences were revised and corrected accordingly and the term “the” or “to” were added in the corresponding lines

  1. References – review and update some of the references where the web page is not provided e.g. Refs 2 and 14.

Response: We thank the reviewer for this observation. The references 2 and 14 were updated

  1. Abstract: conclusion needs to be rephrased. States “… opportunities for three options:…” but three options are not given? Meaning is unclear.

Response: The sentence was revised and the term “three options” was deleted to make the sentence clear

  1. Introduction: Line 44/45 – suggest giving the number of GAVI-eligible countries in AFRO upfront as the focus of the article is on this region. Include details of the different Indian vaccine formulations. 

Response: We thank the reviewer for this suggestion, the details of the different Indian vaccine formulations was added in the revised version of the manuscript

  1. Methods: The study is only descriptive. I did not see any frequencies in the tables. The methods should state this rather than stating that “analysis performed using stata version 14 (StataCorp LP, College Station, Texas, USA), and summarized in tables of frequencies.” (line 89,90).

Response: We thank the reviewer for this comments. The methods section was reviewed and corrected as appropriate

  1. Results:
  2. Table 1: Text states that Nigeria adopted the recently pre-qualified Indian manufactured vaccines (line 103,104) but Table 1 shows that Nigeria introduced Rotarix (RV1). Please clarify.

Response: We apologies for the mistake, and we would like to clarify that Nigeria adopted the Indian manufactured vaccine. We removed Nigeria in the table 2 as the focus of this table are countries that have switched the initially adopted rotavirus products.

  1. Table 2: Line 106 states eight countries have switched to an alternate vaccine yet Table 2 shows 9 countries. Text states that Benin adopted the recently pre-qualified Indian manufactured vaccines (line 103,104) but Table 2 shows Benin with date of introduction and date of switch as Dec 2019 with switch from Rotateq to Rotavac. Please clarify.

Response: We apologies for the mistake as Benin initially adopted Rotavac instead of Rotateq. Benin was removed in the table and the number of countries that have switched the rotavirus products are eight. This was corrected in current version of the manuscript

  1. Table 2: Column 3 should be “Date of introduction” rather than “Year of introduction”. Column 4 (Has the country switched vaccine) can be removed as the table only shows countries that have switched.

Response: We thank the reviewer for this suggestion, we removed the column “Has the country switched vaccine” and the “Year of introduction” was changed to the “Date of introduction”

  1. Table 2 is repeated – remove duplicate table.

Response: The duplicated table was removed

  1. Data are not complete for Table 2 - Not all countries have “Reason for switch” and “Date of switch” in the table.  

Response: The missing information was completed in the revised version of the Manuscript

  1. Table 3: Are there price differences for the different Rotavac and RotaSiil formulations? Which formulations have been used in the countries adopting the Indian vaccines?

Response: Based on information available from cost-effectiveness studies, the difference of cost between Rotavac and RotaSiil formulations is $0.10. For the countries that adopted Indian vaccines (n=5), three adopted Rotavac and two RotaSill. This information was added in the text to make it clear.

  1. It would also be good to provide some additional details on the different Indian vaccine formulations and benefits of using one over another as well as availability (either in the introduction section or when describing Table 3 in the results).

Response: Although data on the vaccine formulation and efficacy is available in the literature, we are unable to compare the benefits of using the Indian vaccines against in the routine immunization programs, as the focus of this manuscript was to document the countries that have switched. In addition, were are unable to predict vaccine availability as the manufacturer do not disclosure possible supply issues. Therefore, table 3 was removed as suggested by other reiewers

  1. Discussion:
  2. Line 139-141 discusses factors complicating the switch in vaccines. Can this be shown in the results when presenting reasons for switching. Assuming these data were collected?

Response: We recognized that this information was not presented in the results, it was because this information is available from the programmatic point of view and not for this study specific. The sentence in the discussion was reviewed and amended

  1. Line 149-150: Is it possible to give an indication of supply available for the different vaccines in the results section e.g. in Table 3? From the discussion, the issue seems to be with supplies of Rotarix, but are there adequate supplies of the two Indian vaccines? Or would this require additional manufacturers?

Response: We thank the reviewer for these comments and questions. As table 3 was removed based on the recommendation of other reviewer, we feel that there is no more need to add this in the results. Additionally, with regards to supply issues, we are unable to elaborate more because the manufacturers do not communicate or anticipate the issues on supply.

  1. Conclusion – suggest revising this sentence as is very long and it’s difficult to understand the overall meaning.

Response: The sentence of conclusion was revised and shortened as suggested

Reviewer 2 Report

Manuscript ID: vaccines-2100549

Title: Lessons learned and future perspectives for rotavirus vaccines switch in the World Health Organization, Regional Office for Africa

The paper evaluates the rotavirus vaccine products introduced in the WHO/AFRO region and examines whether African countries have switched products to the Indian manufactured vaccines and the reason for the switch. While the paper has merit it requires a major revision as some of the data reported in the results section has not been discussed (Table 3 Characteristics of rotavirus vaccines) and in the context of this report could be omitted or put in supplementary data. Furthermore, some of the data in the discussion has not been reported in the results section – major factors complicating the switch as the authors are not clear whether this comes from the data they collected or from the literature.

Additional comments:

Line 21: “…the WHO Africa region (WHO/AFRO).”

Line 23: “…to switch vaccine products. The recently WHO pre-qualified vaccines (Rotavac, Rotasiil), manufactured in India, offer alternatives and reduce global supply challenges related to rotavirus vaccines.”

Line 26:  Please move Methods section to a new line

Line 26-27: Please indicate from whom you collected data?

Line 28: “…WHO/AFRO.”

Line 28: Please move Results section to a new line

Line 28: “…, 35 (92%) adopted Rotateq…and 23% (8/35) switched between products after rotavirus vaccine introduction to Rotavac (n=3), …”

Line 30: “… introduced the rotavirus vaccines manufactured in India.”

Line 31: “…to the Indian vaccines …”

Line 34: Please move Conclusions section to a new line

Line 34-38: This section does not make sense – the authors state that the Indian vaccines offer three options but only state 2. Consider including numbering and rewriting this section for more clarity.

Line 54: “However, vaccine cost,…”

Line 57: “Also, local Government co-financing is required to …”

Line 58: “Cost-effectiveness analysis…compared to the competitor products [8],...”

Line 63: “The decisions influencing switching from one vaccine product to another should systematically assess multiple criteria beyond vaccine price and include the health, programmatic and economic aspects to fully evaluate all product characteristics including …”

Line 67: “…to be use in a country should be based on …”

Line 73: “In WHO/AFRO, similarly to Pakistan, some countries have switched from Rotateq or Rotarix to Rotavac or Rotasiil but the reasons for the decisions to switch vaccines are …”

Line 75: “In this evaluation, we investigated the rotavirus vaccines being use in WHO/AFRO countries and assessed reasons for switching vaccine products at a country level.”  In the paper there are no results showing the lessons learned in the adoption of the Indian-manufactured vaccines and so either add this data or delete the aim. There is some discussion lines 139-141 but no indication whether this was data collected from the current survey or obtained from literature.

Line 80: “To assess what rotavirus vaccines countries in WHO/AFRO that participate in the African Rotavirus Surveillance Network introduced in their immunization programs and whether these vaccines had changed since introduction, we collected information using a standardized questionnaire. Data was also collected by literature review or checking the global introduction status.” If you used a website, please provide the reference to it here.

The data collected included the type of vaccine and if the country had changed or planned to change products since introduction.”

Line 85-87: Not sure that is data is relevant to this assessment as it was not analysed in the context of reasons that countries switched or challenges that countries faced when they did switch. Either include an analysis using this data or delete.

Line 93: “…with Nigeria the most recent in a phased introduction from August 2022.”

Line 96: “…GAVI funding for vaccine administration…have graduate or transitioned, meaning that local Governments must find funding for the vaccines. The remaining countries are GAVI Phase III eligible (Table 1).”

Line 104: “…35 introduced Rotateq or Rotarix.”

Line 105: “In 2009, when rotavirus vaccination was being rolled out, only two vaccine products were available (Rotateq and Rotarix).”

Line 106: This is incorrect. In Table 2, only 5 countries have switched from Rotateq/Rotarix to Rotavac/Rotasill. It looks like Benin originally wanted to introduce Rotateq but didn’t due to supply issues so introduced Rotavac – I would argue that they didn’t switch as Rotateq was never introduced. The remaining 3 countries switched from Rotateq to Rotarix. Either amend sentence to indicate that seven of the 35 countries switched vaccine products after introduction or that five of the 35 countries switched to Indian-manufactured products after introduction.

Line 113: “…switch due to supply issues. Ghana’s decision to switch vaccine products was due to cost as the country graduated from GAVI support and had to self-fund the vaccine program.”

Line 119 – two copies of table 2 in the paper, please delete one copy.

Table 2 – Consider deleting column with question Has country switched vaccines as all countries in this table have switched, Consider coding the reasons for switching as foot notes at the bottom of the table – it may make reading easier. For example,

Country

Vaccine

Date of introduction

Date of switch

New vaccine

Reason for switch#

Benin

Rotateq

N/A

Dec 2019

Rotavac

1

Rwanda

Rotateq

May 2012

Unknown

Rotarix

1, 3

Tanzania

Rotarix

Dec 2012

Oct 2022

Rotavac

4

#Reason for switch code

1 Challenges with supply after Rotateq removed from GAVI market

2 Cost factors associated with graduation from GAVI support

3 Challenges in cold chain

4 Supply shortages

Table 2 also has some data missing – please explain in the table or in the text.

Table 3 is unnecessary in the current paper – delete or include in the analysis.

Line 128: “…was driven by support from GAVI as….WHO/AFRO, did so with…”

Line 133: “…in 2018 [1], offers countries that have…wider vaccine product choices and pricing. In addition, the competitive market safeguards reasonable pricing and aids countries in ensuring long-term sustainability of their vaccine programs after graduation from GAVI support.”

Line 138: Please rewrite this sentence as the authors are implying that 8/35 countries have switched to Indian-manufactured vaccines but this is not the case.

Line 139: Not sure where this data comes from as it was not reported in the results section and there is no reference. Please rectify.

Line 148: “…because of the removal of Rotateq..”

Line 149: “…of rotavirus vaccine production…as some countries are facing supply shortages of their chosen product and switched to the Indian-manufactured vaccines for example Tanzania. ”

Line 152: “As many countries will transition from GAVI-eligibility in future, more countries will have to find additional resources to procure vaccines and fund their vaccine programs potentially necessitating a switch to a lower cost product.”

Line 154-156: This sentence is unclear – please rewrite.

Line 158: “…were notified by GAVI that they should switch to Indian-manufactured vaccines due to unforeseen supply constraints with their current vaccine products.”

Line 168: “…Rotarix vaccine with five African countries currently affected.”

Line 171: “…full GAVI support…”

Author Response

Reviewer 2

  1. The paper evaluates the rotavirus vaccine products introduced in the WHO/AFRO region and examines whether African countries have switched products to the Indian manufactured vaccines and the reason for the switch. While the paper has merit it requires a major revision as some of the data reported in the results section has not been discussed (Table 3 Characteristics of rotavirus vaccines) and in the context of this report could be omitted or put in supplementary data.

Response: We thank the reviewer for this important comment. As per recommendation of other reviewers, table 3 was removed from the revised version of the manuscript

  1. Furthermore, some of the data in the discussion has not been reported in the results section – major factors complicating the switch as the authors are not clear whether this comes from the data they collected or from the literature.

Response: We thank the reviewer for this comment and it was also appointed by other reviewer. This information is available from the programmatic point of view but is not available for this study specific. The sentence in the discussion was modified

Additional comments:

  1. Line 21: “…the WHO Africa region (WHO/AFRO).”

Response: We thank the reviewer for the suggestion, the sentence was corrected

  1. Line 23: “…to switch vaccine products. The recently WHO pre-qualified vaccines (Rotavac, Rotasiil), manufactured in India, offer alternatives and reduce global supply challenges related to rotavirus vaccines.”

Response: The sentence was corrected as recommended

  1. Line 26:  Please move Methods section to a new line

Response: Methods were moved to the new line

  1. Line 26-27: Please indicate from whom you collected data?

Response: This information was added in the methods

  1. Line 28: “…WHO/AFRO.”

Response: Corrected

  1. Line 28: Please move Results section to a new line

Response: Results were moved to the new line

  1. Line 28: “…, 35 (92%) adopted Rotateq…and 23% (8/35) switched between products after rotavirus vaccine introduction to Rotavac (n=3),…”

Response: The sentence was corrected

  1. Line 30: “… introduced the rotavirus vaccines manufactured in India.”

Response: The sentence was corrected

  1. Line 31: “…to the Indian vaccines …”

Response: The sentence was corrected

  1. Line 34: Please move Conclusions section to a new line

Response: Conclusions were moved to the new line

  1. Line 34-38: This section does not make sense – the authors state that the Indian vaccines offer three options but only state 2. Consider including numbering and rewriting this section for more clarity.

Response: We thank for the comment, the sentence was revised and corrected

  1. Line 54: “However, vaccine cost,…”

Response: The sentence was corrected

  1. Line 57: “Also, local Government co-financing is required to …”

Response: The sentence was corrected

  1. Line 58: “Cost-effectiveness analysis…compared to the competitor products [8],...”

Response: The sentence was corrected

  1. Line 63: “The decisions influencing switching from one vaccine product to another should systematically assess multiple criteria beyond vaccine price and include the health, programmatic and economic aspects to fully evaluate all product characteristics including …”

Response: The sentence was corrected

  1. Line 67: “…to be use in a country should be based on …”

Response: The sentence was corrected

  1. Line 73: “In WHO/AFRO, similarly to Pakistan, some countries have switched from Rotateq or Rotarix to Rotavac or Rotasiil but the reasons for the decisions to switch vaccines are …”

Response: The sentence was corrected

  1. Line 75: “In this evaluation, we investigated the rotavirus vaccines being use in WHO/AFRO countries and assessed reasons for switching vaccine products at a country level.”  In the paper there are no results showing the lessons learned in the adoption of the Indian-manufactured vaccines and so either add this data or delete the aim. There is some discussion lines 139-141 but no indication whether this was data collected from the current survey or obtained from literature.

Response: We thank the reviewer for the comment and the sentence was revised and corrected

  1. Line 80: “To assess what rotavirus vaccines countries in WHO/AFRO that participate in the African Rotavirus Surveillance Network introduced in their immunization programs and whether these vaccines had changed since introduction, we collected information using a standardized questionnaire. Data was also collected by literature review or checking the global introduction status.” If you used a website, please provide the reference to it here.

The data collected included the type of vaccine and if the country had changed or planned to change products since introduction.”

Response: The sentence was corrected

  1. Line 85-87: Not sure that is data is relevant to this assessment as it was not analyzed in the context of reasons that countries switched or challenges that countries faced when they did switch. Either include an analysis using this data or delete.

Response: We thank the reviewer for the comment and this information was deleted in the revised version of the manuscript

  1. Line 93: “…with Nigeria the most recent in a phased introduction from August 2022.”

Response: The sentence was corrected

  1. Line 96: “…GAVI funding for vaccine administration…have graduate or transitioned, meaning that local Governments must find funding for the vaccines. The remaining countries are GAVI Phase III eligible (Table 1).”

Response: We thank the reviewer for the suggestion and the sentence was reviewed and corrected as appropriate

  1. Line 104: “…35 introduced Rotateq or Rotarix.”

Response: The sentence was corrected

  1. Line 105: “In 2009, when rotavirus vaccination was being rolled out, only two vaccine products were available (Rotateq and Rotarix).”

Response: The sentence was corrected

  1. Line 106: This is incorrect. In Table 2, only 5 countries have switched from Rotateq/Rotarix to Rotavac/Rotasill. It looks like Benin originally wanted to introduce Rotateq but didn’t due to supply issues so introduced Rotavac – I would argue that they didn’t switch as Rotateq was never introduced. The remaining 3 countries switched from Rotateq to Rotarix. Either amend sentence to indicate that seven of the 35 countries switched vaccine products after introduction or that five of the 35 countries switched to Indian-manufactured products after introduction.

Response: We thank the reviewer for the comment. The sentence was revised and corrected

  1. Line 113: “…switch due to supply issues. Ghana’s decision to switch vaccine products was due to cost as the country graduated from GAVI support and had to self-fund the vaccine program.”

Response: The sentence was corrected accordingly

  1. Line 119 – two copies of table 2 in the paper, please delete one copy.

Response: One table was deleted

  1. Table 2 – Consider deleting column with question Has country switched vaccines as all countries in this table have switched, Consider coding the reasons for switching as foot notes at the bottom of the table – it may make reading easier. For example,

Country

Vaccine

Date of introduction

Date of switch

New vaccine

Reason for switch#

Benin

Rotateq

N/A

Dec 2019

Rotavac

1

Rwanda

Rotateq

May 2012

Unknown

Rotarix

1, 3

Tanzania

Rotarix

Dec 2012

Oct 2022

Rotavac

4

#Reason for switch code

1 Challenges with supply after Rotateq removed from GAVI market

2 Cost factors associated with graduation from GAVI support

3 Challenges in cold chain

4 Supply shortages

Response: The column with question has country switched vaccines was deleted and the coding of reasons for switch code was added as footnote

  1. Table 2 also has some data missing – please explain in the table or in the text.

Response: The missing information was completed

  1. Table 3 is unnecessary in the current paper – delete or include in the analysis.

Response: As suggested by other reviewers, table 3 deleted

  1. Line 128: “…was driven by support from GAVI as….WHO/AFRO, did so with…”

Response: The sentence was corrected

  1. Line 133: “…in 2018 [1], offers countries that have…wider vaccine product choices and pricing. In addition, the competitive market safeguards reasonable pricing and aids countries in ensuring long-term sustainability of their vaccine programs after graduation from GAVI support.”

Response: The sentence was corrected

  1. Line 138: Please rewrite this sentence as the authors are implying that 8/35 countries have switched to Indian-manufactured vaccines but this is not the case.

Response: The sentence was re-written

  1. Line 139: Not sure where this data comes from as it was not reported in the results section and there is no reference. Please rectify.

Response: The sentence was revised and corrected to clarify

  1. Line 148: “…because of the removal of Rotateq..”

Response: The sentence was corrected

  1. Line 149: “…of rotavirus vaccine production…as some countries are facing supply shortages of their chosen product and switched to the Indian-manufactured vaccines for example Tanzania. ”

Response: The sentence was corrected

  1. Line 152: “As many countries will transition from GAVI-eligibility in future, more countries will have to find additional resources to procure vaccines and fund their vaccine programs potentially necessitating a switch to a lower cost product.”

Response: The sentence was corrected

  1. Line 154-156: This sentence is unclear – please rewrite.

Response: The sentence was revised and corrected

  1. Line 158: “…were notified by GAVI that they should switch to Indian-manufactured vaccines due to unforeseen supply constraints with their current vaccine products.”

Response: The sentence was corrected

  1. Line 168: “…Rotarix vaccine with five African countries currently affected.”

Response: The sentence was corrected

  1. Line 171: “…full GAVI support…”

Response: The sentence was corrected

Reviewer 3 Report

Lessons learned and future perspectives for rotavirus vaccines switch in the World Health Organization, Regional Office for Africa

By I Mandomando et al (Corresponding author: JM Mwenda)

Submitted to Vaccines(Editorial No vaccines-2100549)

General Comments

This is a review of the present situation of the introduction of rotavirus vaccination into EPIs of African countries. Due to cost, supply and other issues, several countries had to switch supplier or are planning to do so. As far as reviewer can see this is an up-to-date report which is important in the context of children’s health in Africa and worldwide. A few requests for clarification are made, and Table 2 requires a thorough revision. 

Specific Comments

Line 

16        Change superscript to ‘7’. 

35        … three options… Those should be formulated more clearly. 

54        Consider citation of: 

Soares-Weiser K, Bergman H, Henschke N, Pitan F, Cunliffe N. Vaccines for preventing rotavirus diarrhoea: vaccines in use. Cochrane Database Syst Rev. 2019 Oct 28;2019(10):CD008521. doi: 10.1002/14651858.CD008521.pub5. Update in: Cochrane Database Syst Rev. 2021 Nov 17;11:CD008521. PMID: 31684685; PMCID: PMC6816010.

Bergman H, Henschke N, Hungerford D, Pitan F, Ndwandwe D, Cunliffe N, Soares-Weiser K. Vaccines for preventing rotavirus diarrhoea: vaccines in use. Cochrane Database Syst Rev. 2021 Nov 17;11(11):CD008521. doi: 10.1002/14651858.CD008521.pub6. PMID: 34788488; PMCID: PMC8597890.

Henschke N, Bergman H, Hungerford D, Cunliffe NA, Grais RF, Kang G, Parashar UD, Wang SA, Neuzil KM. The efficacy and safety of rotavirus vaccines in countries in Africa and Asia with high child mortality. Vaccine. 2022 Mar 15;40(12):1707-1711. doi: 10.1016/j.vaccine.2022.02.003. Epub 2022 Feb 17. PMID: 35184924; PMCID: PMC8914343.

The ref. Henschke et al is particularly relevant in the context of this submission. 

68        … country, either at first introduction or at switch time, should be based… 

88        … Data were collected… 

94        … being the most recent country that… 

96        Summarize the reasons for which the 6 names countries were not eligible for GAVI funding.

109      Table 2. The table contains numerous duplications, mainly in the second half (line 119ff). The source of data column should be omitted. A thorough revision of the table is required. 

138      … including one that has graduated… 

154      to 156. Please rephrase sentence for clarification. 

171      … these challenges emphasize the need  of… 

185      Enter details of authors’ contributions. 

192f     References

            Refs 2, 5, and 14 are incomplete. 

Author Response

Reviewer 3

General Comments

  1. This is a review of the present situation of the introduction of rotavirus vaccination into EPIs of African countries. Due to cost, supply and other issues, several countries had to switch supplier or are planning to do so. As far as reviewer can see this is an up-to-date report which is important in the context of children’s health in Africa and worldwide. A few requests for clarification are made, and Table 2 requires a thorough revision. 

Response: Table 2 was reviewed e corrected

Specific Comments

Line 

  1. Line 16        Change superscript to ‘7’. 

Response: The superscript was changed to 7

  1. Line 35        … three options… Those should be formulated more clearly. 

Response: The sentence related to the options was revised and modified n the new version of the manuscript

  1. Line 54        Consider citation of: 

Soares-Weiser K, Bergman H, Henschke N, Pitan F, Cunliffe N. Vaccines for preventing rotavirus diarrhoea: vaccines in use. Cochrane Database Syst Rev. 2019 Oct 28;2019(10):CD008521. doi: 10.1002/14651858.CD008521.pub5. Update in: Cochrane Database Syst Rev. 2021 Nov 17;11:CD008521. PMID: 31684685; PMCID: PMC6816010.

Bergman H, Henschke N, Hungerford D, Pitan F, Ndwandwe D, Cunliffe N, Soares-Weiser K. Vaccines for preventing rotavirus diarrhoea: vaccines in use. Cochrane Database Syst Rev. 2021 Nov 17;11(11):CD008521. doi: 10.1002/14651858.CD008521.pub6. PMID: 34788488; PMCID: PMC8597890.

Henschke N, Bergman H, Hungerford D, Cunliffe NA, Grais RF, Kang G, Parashar UD, Wang SA, Neuzil KM. The efficacy and safety of rotavirus vaccines in countries in Africa and Asia with high child mortality. Vaccine. 2022 Mar 15;40(12):1707-1711. doi: 10.1016/j.vaccine.2022.02.003. Epub 2022 Feb 17. PMID: 35184924; PMCID: PMC8914343.

The ref. Henschke et al is particularly relevant in the context of this submission. 

Response: We thank the reviewer for this suggestion which was taken into consideration 

  1. Line 68        … country, either at first introduction or at switch time, should be based… 

Response: The sentence was corrected in the revised version

  1. Line 88        … Data were collected… 

Response: The sentence was corrected

  1. Line 94        … being the most recent country that… 

Response: The sentence was corrected

  1. Line 96        Summarize the reasons for which the 6 names countries were not eligible for GAVI funding.

Response: The reasons for which the 6 names countries were not eligible for GAVI funding was added in the revised version of the manuscript

  1. Line 109      Table 2. The table contains numerous duplications, mainly in the second half (line 119ff). The source of data column should be omitted. A thorough revision of the table is required. 

Response: Table 2 was revised and modified as recommended also by other reviewers

  1. Line 138      … including one that has graduated… 

Response: The sentence was modified as also suggested by other reviewers

  1. Line 154      to 156. Please rephrase sentence for clarification. 

Response: The sentence was revised and modified as also recommended by reviewers

  1. Line 171      … these challenges emphasize the need of… 

Response: The sentence was corrected

  1. Line 185      Enter details of authors’ contributions. 

Response: The author’s contributions was added in the revised version of the manuscript

  1. Line 192f     References. Refs 2, 5, and 14 are incomplete. 

Response: The references 2, 5, and 14 were reviewed and adjusted accordingly

Reviewer 4 Report

This manuscript describes recent situation of the use of rotavirus vaccines in African countries, particularly after the recent WHO's prequalification. Although this manuscript may be meaningful to understand the use and its trend of rotavirus vaccine in Africa, there is no scientific descriptions without no scientific methodology. Concerns of this manuscript is as follows.

1. The switch of rotavirus vaccines in African countries were described in relation to GAVI. However, in this regard, no information about vaccine efficacy in individual countries was shown. In addition, it should have been better to get information of vaccination rate in each country.

2. Regarding rotavirus vaccines, no medical (virological, immunological, clinical) descriptions were included. There is only description of RV1 and RV5 as vaccine type, but there is no explanation.

3. Method section: Authors did not clearly write how they collected information from African countries. To whom did authors send the questionnaires? When and how long duration did authors collect the information by questionnaire? Otherwise, how did authors perform literature review? What literatures were used? Were the obtained information correct and reliable? How can authors ensure the reliability and correctness of information?    

Author Response

Reviewer 4

65. This manuscript describes recent situation of the use of rotavirus vaccines in African countries, particularly after the recent WHO's prequalification. Although this manuscript may be meaningful to understand the use and its trend of rotavirus vaccine in Africa, there is no scientific descriptions without no scientific methodology. Concerns of this manuscript is as follows.

Response: We thank the reviewer for the comment. The methods were revised and modified

66. The switch of rotavirus vaccines in African countries were described in relation to GAVI. However, in this regard, no information about vaccine efficacy in individual countries was shown. In addition, it should have been better to get information of vaccination rate in each country.

Response: We thank the reviewer for this interesting comment. Data on vaccine efficacy in African countries are available only for RotaSiil, on the study conducted in Niger (PLoS Med 2021; 18(7):e1003655. doi: 10.1371/journal.pmed.1003655) and no data is available for Rotavac. Data on vaccination rates are available from WHO / UNICEF estimates, but we did not include in this manuscript as our focus was vaccine switch from the initially adopted products.

67. Regarding rotavirus vaccines, no medical (virological, immunological, clinical) descriptions were included. There is only description of RV1 and RV5 as vaccine type, but there is no explanation.

Response: Again we thank the reviewer for this comment and as per response above, our main focus was to document the countries that had switched the rotavirus products from the initially introduced.

68. Method section: Authors did not clearly write how they collected information from African countries. To whom did authors send the questionnaires? When and how long duration did authors collect the information by questionnaire? Otherwise, how did authors perform literature review? What literatures were used? Were the obtained information correct and reliable? How can authors ensure the reliability and correctness of information?    

Response: We thank the reviewer for this important questions and comments. As per response of question 67, our may focused was to assess the countries that had switch the vaccine; then we collected data through the country immunization programs managers. The methods were revised and modified for clarity.

Round 2

Reviewer 1 Report

Thank you for addressing my previous comments. I have a few remaining comments/areas for clarification, particularly in the methods section and for Table 1.

1.       Grammar still needs some improvement. There are several sentences that need improvement in sentence structure and/or use of plurals, etc.

2.       Methods:

a.       The abstract mentions that a questionnaire was provided to EPI program managers but this is not detailed in the methods section.

b.       Are all countries in the AFRO region part of the RV surveillance network? Were any countries not sent a questionnaire because they weren’t part of the network. Please clarify.

c.       What is meant by “standard” questionnaire? How were the data collected? Was it an online survey? What was the response rate? Did all countries complete the questionnaire?

d.       Details of the literature review should be provided e.g. which databases, language, search terms, dates, etc.

3.       Results:

a.       Table 1 still shows that Nigeria introduced Rotarix (RV1), not an Indian vaccine. Please amend.

b.       Is Table 1 showing the current vaccine in use in the country or the initial vaccine introduced? This needs to be clear in the Table title/column heading. Please check for errors in the table as there isn’t consistency for the countries that switched. E.g. Ghana – shows vaccine as Rotavac (switch), not Rotarix (initially introduced) but Tanzania shows Rotarix (initially introduced). Rwanda shows Rotarix (switch) not Rotateq (initially introduced); Mali shows the vaccine initially introduced – Rotateq. This is very confusing for the reader.

4.       Discussion:

a.       Line 150/151 – states “personal communication from EPI managers” but abstract states that the questionnaire was sent to the EPI managers. So these data were formally analysed as part of the study, rather than personally communicated. Clarification needed in methods section to address this.

Author Response

Thank you for addressing my previous comments. I have a few remaining comments/areas for clarification, particularly in the methods section and for Table 1.

  1. Grammar still needs some There are several sentences that need improvement in sentence structure and/or use of plurals, etc.

Response: The manuscript has been reviewed by an English speaking native

  1. Methods:
  2. The abstract mentions that a questionnaire was provided to EPI program managers but this is not detailed in the methods section.

Response: Details on the questionnaire administration was expanded in the methods section

  1. Are all countries in the AFRO region part of the RV surveillance network? Were any countries not sent a questionnaire because they weren’t part of the network. Please clarify.

Response: We thank the reviewer for the comment and we would like to clarify that the questionnaire was administered only for the countries that switched or were planning to switch the vaccine product. This was added in the methods section

  1. What is meant by “standard” questionnaire? How were the data collected? Was it an online survey? What was the response rate? Did all countries complete the questionnaire?

Response: For clarity the term ”standard” was removed. The questionnaire were sent by email to the EPI managers and they fill the questionnaire and return to the investigators. The questionnaire was administered only for the countries that had switched or were planning to switch the products, and this was added in the new version of the manuscript  

  1. Details of the literature review should be provided e.g. which databases, language, search terms, dates, etc.

Response: Detals on literature review was added in the method section

  1. Results:
  2. Table 1 still shows that Nigeria introduced Rotarix (RV1), not an Indian vaccine. Please amend.

Response: We thank the reviewer and this was amended in the new version of the manuscript

  1. Is Table 1 showing the current vaccine in use in the country or the initial vaccine introduced? This needs to be clear in the Table title/column heading. Please check for errors in the table as there isn’t consistency for the countries that switched. E.g. Ghana – shows vaccine as Rotavac (switch), not Rotarix (initially introduced) but Tanzania shows Rotarix (initially introduced). Rwanda shows Rotarix (switch) not Rotateq (initially introduced); Mali shows the vaccine initially introduced – Rotateq. This is very confusing for the reader.

Response: We would like to clarify that this table shows the initially adopted rotavirus vaccine products. The title of the table was amended to make it clear

  1. Discussion:
  2. Line 150/151 – states “personal communication from EPI managers” but abstract states that the questionnaire was sent to the EPI managers. So these data were formally analysed as part of the study, rather than personally communicated. Clarification needed in methods section to address this.

Response: We thank the reviewer for the comment, in fact this information is already included in Table 2 and the sentence was corrected as highlighted in the revised version

Reviewer 2 Report

Overall, the authors have improved the article by implementing the suggested corrections. 

Line 162 - 163: This sentence is very unclear and should be revised for clarity "Notably, previous rotavirus vaccine switch that necessitated by factors mentioned earlier that informed decision to switch vaccine, meant country had adequate time to plan and manage the switch." 

Author Response

  1. Overall, the authors have improved the article by implementing the suggested corrections. 

Line 162 - 163: This sentence is very unclear and should be revised for clarity "Notably, previous rotavirus vaccine switch that necessitated by factors mentioned earlier that informed decision to switch vaccine, meant country had adequate time to plan and manage the switch." 

Response: This sentence was reviewed

Reviewer 4 Report

In the method section, authors still write "we collected information using a standard questionnaire.". There is no explanation about the "standard questionnaire", and how and to whom the information  was inquired.

Author Response

  1. In the method section, authors still write "we collected information using a standard questionnaire.". There is no explanation about the "standard questionnaire", and how and to whom the information was inquired.

Response: We thank the reviewer for this comment, we amended the method section in the new version of the manuscript

Round 3

Reviewer 4 Report

Line 85-87: "by searching the terms or combination of the terms “rotavirus vaccine switch in WHO AFRO” in the Pubmed, WHO website" This sentence is still not complete and grammatically not correct. Revise again. In adddition, URL of websites should be added. "Pubmed" should be "PubMed".

Author Response

We the reviewer and sentence has been reviewed in the current version
